# The Level of Conus Medullaris in 629 Healthy Japanese Individuals

**DOI:** 10.3390/jcm10143182

**Published:** 2021-07-19

**Authors:** Hiroaki Nakashima, Keigo Ito, Yoshito Katayama, Mikito Tsushima, Kei Ando, Kazuyoshi Kobayashi, Masaaki Machino, Sadayuki Ito, Hiroyuki Koshimizu, Naoki Segi, Hiroyuki Tomita, Shiro Imagama

**Affiliations:** 1Department of Orthopaedic Surgery, Nagoya University Graduate School of Medicine, Nagoya 466-8560, Japan; andokei@med.nagoya-u.ac.jp (K.A.); k_koba1@f2.dion.ne.jp (K.K.); masaaki_machino_5445_2@yahoo.co.jp (M.M.); sadaito@med.nagoya-u.ac.jp (S.I.); love_derika@yahoo.co.jp (H.K.); naoki.s.n@gmail.com (N.S.); hiro_tomi_1031@yahoo.co.jp (H.T.); imagama@med.nagoya-u.ac.jp (S.I.); 2Department of Orthopedic Surgery, Chubu Rosai Hospital, 1-10-6 Komei, Minato-ku, Nagoya 455-8530, Japan; spine.ort@chubuh.johas.go.jp (K.I.); yokatayama@hotmail.com (Y.K.); meikeihan@hotmail.com (M.T.)

**Keywords:** conus medullaris, height, pelvic incidence, magnetic resonance imaging, healthy volunteers

## Abstract

The conus medullaris typically terminates at the L1 level; however, variations in its level and the factors associated with the conus medullaris level are unclear. We investigated the level of conus medullaris on magnetic resonance imaging in healthy volunteers. In total, 629 healthy adult volunteers (≥50 individuals of each sex and in each decade of age from 20 to 70) were enrolled. The level of the conus medullaris was assessed based on the T2-weighted sagittal magnetic resonance images, and factors affecting its level were investigated employing multivariate regression analysis including the participants’ background and radiographical parameters. L1 was the most common conus medullaris level. Participant height was significantly shorter in the caudally placed conus medullaris (*p* = 0.013). With respect to the radiographical parameters, pelvic incidence (*p* = 0.003), and pelvic tilt (*p* = 0.03) were significantly smaller in participants with a caudally placed conus medullaris. Multiple regression analysis showed that the pelvic incidence (*p* < 0.0001) and height (*p* < 0.0001) were significant factors affecting the conus medullaris level. These results indicated that the length of the spinal cord varies little among individuals and that skeletal differences affect the level of the conus medullaris.

## 1. Introduction

The conus medullaris is located at the terminal end of the spinal cord. The lower-most tapering extremity of the spinal cord is called the conus medullaris [1,2,3,4,5,6]. The thoracolumbar junction includes the conus medullaris and cauda equina. Injury to these neurological structures is associated with functional consequences. The conus medullaris and cauda equina are a transition point from the central to the peripheral nervous system, and injury to this point can result in a series of upper and lower motor neuron symptoms, depending on the location of the injury.

Although its level varies between T12 and lower L2, it typically lies at the inferior aspect of the L1 vertebra in adults [1,2,3,4,5,6]. The level of the conus medullaris is important in spinal anesthesia and spinal surgeries. However, few studies have investigated the factors affecting the conus medullaris level; sex, and age have been reported as potential factors [5,6,7,8]. With respect to children, the conus medullaris is placed caudally to L2 vertebrae in children younger than 1 year of age; however, it is found in the lower third of L1 after 1 year of age [6,8]. There remains controversy as to whether age affects the conus medullaris level in adults [9], and the influence of sex is also controversial.

The problems with previous studies are that (1) few large-scale studies involving older adults have been performed employing magnetic resonance imaging (MRI); (2) few studies have investigated the physical aspect of the participants, such as height and weight; and (3) there are no reports on the relationship between spinal alignment on X-ray photographs (Xp) and conus medullaris on MRI. Also, the significant factors affecting the level of the conus medullaris in adults are unclear. The aim of the current study was to investigate the levels of conus medullaris on MRI in healthy individuals and identify the factors that determine the conus medullaris location, including body size and radiographical spinal alignment. 

## 2. Materials and Methods

### 2.1. Study Participants

Japanese volunteers were prospectively recruited after the purpose of this study was officially announced and after obtaining institutional review board approval from the Chubu Rosai Hospital (IRB approval no., 2009-2). Written informed consent was obtained from all participants. As part of a comprehensive medical examination, the study was conducted after consent was obtained from subjects who wanted spinal examinations. Participants were offered free feedback on findings from spine radiographs and MRIs, rather than monetary rewards. All of the included volunteers understood the negative effects of radiation exposure and agreed to undergo an X-ray examination. We prospectively recruited the subjects using newspaper advertisements and posters in facilities having some sort of relationship with our hospital. The majority of the subjects were not patients at our hospital but relatively healthy residents of the area. This study was registered in the research database at the Rosai Hospital in Japan.

The exclusion criteria included a history of brain or spinal surgery; comorbid neurological disease, such as cerebral infarction or neuropathy; symptoms related to sensory or motor disorders (numbness, clumsiness, motor weakness, or gait disturbances); intermittent claudication; and severe low back pain. Visual analogue scale (VAS) measurements of the lower back, buttock, and leg pain were taken before deciding on the inclusion of patients in this study and excluded cases with severe pain anywhere above 80 mm as cases with severe pain. Pregnant women and individuals who received worker’s compensation or who presented with symptoms after a motor vehicle accident were also excluded. If radiographic measurements of the sagittal parameters were difficult to assess due to lumbosacral transitional anomalies, the participants were also excluded. We also excluded cases with a previous medical history of vertebral fracture, spinal infection, rheumatoid arthritis, autoimmune diseases, or chronic renal failure. In contrast, we included cases with diabetes mellitus or smoking history. Finally, 629 individuals with appropriate images were enrolled: the study population included at least 50 participants of each sex and each decade of age from 20 to 70. The study included 308 men (50 in their 20s, 51 in their 30s, 50 in their 40s, 56 in their 50s, 51 in their 60s, and 50 in their 70s) and 321 women (53 in their 20s, 50 in their 30s, 57 in their 40s, 51 in their 50s, 60 in their 60s, and 50 in their 70s).

### 2.2. Radiographical Examinations

We performed MRI scans on a 1.5-Tesla superconducting magnet (Signa Horizon Excite HD version 12; GE Healthcare, UK). Scans were taken at slice thicknesses of 3 mm in the respective sagittal planes. We obtained T1-weighted images (fast spin-echo repetition time (TR), 450 ms; echo time (TE), 13 ms), and T2-weighted images (fast spin-echo TR, 4000 ms; TE, 85 ms). All images were transferred to the computer as Digital Imaging and Communications in Medicine (DICOM) data. The tip of the conus medullaris can be identified on midline sagittal T1- and T2-weighted MRI.

Furthermore, full-length, free-standing spinal radiographs with fists on the clavicles were obtained from all the participants. All the images were transferred to a computer as DICOM data. The sagittal vertical axis (SVA), cervical lordosis, thoracic kyphosis, lumbar lordosis (LL), pelvic incidence (PI), and pelvic tilt (PT) were measured. Each parameter was manually measured by experienced radiation technologists (single measurements by random raters) under the supervision of a certified spine surgeon, using imaging software (Osiris version4; Icestar Media Ltd., Essex, UK). 

### 2.3. Statistical Analysis

Each variable was reported as the mean ± standard deviation. At first, we assessed the standard distribution of each parameter (age, height, weight, BMI, and radiographical parameters using the Kolmogorov-Smirnov test. After confirmation of the normal distribution, we employed the one-way ANOVA (post hoc Tukey) to investigate the differences for each parameter at the different conus medullaris levels. The Chi-square test was used for testing relationships between categorical variables. In addition, a multivariate regression analysis was performed to determine the significant contributory factors at each level of the conus medullaris. We employed the step-wise method for the multivariate regression analysis and included factors with a *p*-value of <0.05. *p*-values of <0.05 were considered to be indicative of statistical significance. All analyses were performed with the IBM SPSS Statistics for Windows, Version 27.0 (IBM Corp., Armonk, NY, USA).

## 3. Results

The conus medullaris level was Th11-12, T12, T12-L1, L1, L1-2, and L2 in 3 (0.5%), 46 (7.3%), 204 (32.4%), 288 (45.8%), 79 (12.6%), and 9 (1.4%) participants, respectively, and L1 was the most common level.

Next, we investigated the effect of the physique on the level of the conus medullaris. The participants’ heights were significantly shorter in the caudally placed conus medullaris cases (163.7, 163.9, 163.2, 162.7, 159.5, and 157.4 cm in the Th11-12, T12, T12-L1, L1, L1-2, and L2 conus medullaris levels, respectively; *p* = 0.013). On the other hand, there were no significant differences related to gender (*p* = 0.48), body weight (*p* = 0.14) or body mass index (BMI) (*p* = 0.96) (Table 1). Age was also not significantly different among the conus medullaris levels (*p* = 0.86 in Table 1). 

With respect to the relationship between the radiographical parameters and the conus medullaris levels, PI (62.0°, 58.0°, 55.0°, 52.7°, 50.9°, and 49.6° in the Th11-12, T12, T12-L1, L1, L1-2, and L2 conus medullaris level, respectively; *p* = 0.003) and PT (18.4°, 18.2°, 15.7°, 13.9°, 14.6°, and 12.4° in the Th11-12, T12, T12-L1, L1, L1-2, and L2 conus medullaris level, respectively; *p* = 0.03) were significantly smaller in the participants with caudal cauda equina (Table 2). The LL was smaller in the caudal levels of conus medullaris (56.7°, 52.1°, 50.5°, 49.1°, 46.9°, and 46.0° in the Th11-12, T12, T12-L1, L1, L1-2, and L2 conus medullaris level, respectively; *p* = 0.10), although the difference did not reach statistical significance. However, there were no significant differences in cervical lordosis, thoracic kyphosis, or SVA (Table 2).

In order to analyze the data in further detail, height, PI, and PT were divided into categories and examined again (Table 3). With respect to the PI, there was a significant difference (*p* = 0.045) when the conus medullaris was located in the cranial side in the case of high PI, but there was no significant difference in the cases of other heights (*p* = 0.67) and PT (*p* = 0.12).

Multiple regression analysis showed that PI (standardized β coefficient: −0.18, *p* < 0.0001) and height (standardized β coefficient: −0.16, *p* < 0.0001) were significant factors affecting the level of the conus medullaris, although age, sex, weight, BMI, and other radiographical parameters were not significant.

## 4. Discussion

This study investigated the anatomical level of the conus medullaris and analyzed factors associated with the conus medullaris levels in 629 healthy volunteers. In the present study, the majority (92.2%) of the participants had the conus medullaris at the caudal level of the T12-L1 disk, and the conus medullaris was located cranially to the T12 vertebral level in only 7.8% of the participants. Among them, the T12-L1 disk and L1 vertebral body were the most common conus medullaris levels, which were 32.4% and 45.8%, respectively.

Our study demonstrated that shorter height and smaller PI were significantly associated with a caudally placed conus medullaris. This result might indicate that the length of the spinal cord varies little among individuals and that the skeletal difference affects the conus medullaris level. In addition to height, PI was a key driver of the conus medullaris level. Individuals with a larger PI typically have greater LL and thoracic kyphosis, and the end of the spinal cord might be located more cranially in the twisted spinal canal. However, as far as we know, there is no paper showing the relationship between the PI and the conus level due to the lack of studies investigating the conus level by using both lumbar MRI and X-rays. For this reason, the current results will need to be verified in future studies.

The location of the conus medullaris varies by developmental stage [6,8]. At birth, the cord fills the vertebral canal and terminates at the lumbosacral junction [8]. The distal end of the spinal cord then moves toward the cranial direction with infant development [6,8], probably because of the differential growth between the spinal column and spinal cord. In adults, the tip usually terminates at the mid aspect of the L1 vertebra. However, its position varies between the lower 11th thoracic and upper third lumbar vertebrae [5]. In a cadaveric study, the spinal cord measured roughly 45 cm in the adult male and 42 cm in the adult female [10]. The current results might indicate that the variation in spinal cord length is limited, and the skeletal anatomy of height and spino-pelvic sagittal alignment varies among individuals. 

PI is one of the most important radiographical parameters in the case of spinal sagittal alignment [11]. The PI increases during childhood as the spine adapts to bipedal walking and stabilizes after adulthood [12]. PI strongly correlates with LL through the sacral slope (SS), and the larger PI is associated with a larger LL. Despite its great importance, PI varies from 33° to 85° among adults [13] and largely affects spinal sagittal alignment. Recent retrospective studies suggested that distal LL (L4-S1) is comparable between low to moderate and high PI groups. Proximal LL (L1-L4), however, is significantly influenced by the PI value (greater PI, and greater proximal lumbar lordosis) [14,15]. Furthermore, not only does the LL magnitude increase in cases of a larger PI but also the LL apex and inflection point are located more toward the cranial side [14]. Thus, in cases with a large PI the local lordosis around L1, where the conus medullaris is often located [1,2,3,4,5,6], might be greater, and the conus medullaris might be located more toward the cranial side in the twisted spinal canal. The present study did not measure local sagittal alignment around L1, and so this discussion is only speculative. The relationship between PI and conus medullaris needs to be further investigated.

### Strengths and Limitations

A strength of this study was that it was a relatively large-scale study including ≥50 individuals of each sex and decade of age (20s–70s). Furthermore, both MRI and Xp were obtained in all subjects. As a limitation of the current study, the participants were a single race of Japanese. This limitation might affect the size and place of the spinal column and spinal cord. An international large-scale multicenter study is warranted to validate our results. As a second limitation, cases with lumbosacral transitional anomalies were excluded in the current study, however, it is necessary to examine the level of conus in these cases of transitional vertebra in the future. Lastly, we could not compare spinal alignment and the level of conus medullaris by degrees of pain, although the degrees of pain might affect the results. Future detailed studies assessing the pain are needed.

## 5. Conclusions

The majority of participants had the conus medullaris at the caudal level of the T12-L1 disk (92.2%), and the conus medullaris was located cranially to the T12 vertebral level in only 7.8% of the 629 healthy volunteers. Lower height and smaller PI were associated with the caudally placed conus medullaris; thus, skeletal differences were significantly associated with the conus medullaris level.

## Figures and Tables

**Table 1 jcm-10-03182-t001:** The association between the conus medullaris level and patients’ backgrounds.

	T11-12	T12	T12-L1	L1	L1-2	L2	*p*
age (yr)	39.7 ± 15.0	49.2 ± 15.5	49.0 ± 15.9	50.3 ± 17.0	49.6 ± 16.2	48.2 ± 23.4	0.86
gender (male/female)	0/3	20/26	96/108	146/142	41/38	5/4	0.48
height (cm)	163.7 ± 6.8	163.9 ± 9.8	163.2 ± 8.9	162.7 ± 8.7	159.5 ± 8.9	157.4 ± 10.8	0.013
body weight (kg)	60.7 ± 8.1	60.3 ± 12.1	60.6 ± 12.4	59.9 ± 10.6	56.8 ± 12.1	54.3 ± 6.7	0.14
BMI	22.7 ± 2.7	22.3 ± 3.5	22.6 ± 3.4	22.6 ± 3.1	22.2 ± 3.7	22.0 ± 2.4	0.97

yr: years of age, BMI: body mass index.

**Table 2 jcm-10-03182-t002:** The association between the conus medullaris level and radiographical parameters.

	T11-12	T12	T12-L1	L1	L1-2	L2	*p*
CL (°)	8.7 ± 8.5	4.0 ± 15.0	3.3 ± 10.8	4.5 ± 12.5	3.9 ± 11.7	12.6 ± 9.1	0.29
TK (°)	43.7 ± 8.3	32.8 ± 14.9	32.5 ± 19.7	34.9 ± 14.6	30.0 ± 18.6	36.2 ± 6.0	0.19
LL (°)	56.7 ± 11.4	52.1 ± 11.5	50.5 ± 11.9	49.1 ± 11.9	46.9 ± 14.3	46.0 ± 8.1	0.10
PI (°)	62.0 ± 12.2	58.0 ± 13.5	55.0 ± 12.0	52.7 ± 11.0	50.9 ± 11.5	49.6 ± 9.6	0.003
PT (°)	18.4 ± 7.8	18.2 ± 9.5	15.7 ± 9.0	13.9 ± 8.2	14.6 ± 10.6	12.4 ± 7.8	0.03
SVA (cm)	2.1 ± 4.9	3.0 ± 5.0	1.2 ± 5.4	1.8 ± 5.5	1.9 ± 6.5	1.7 ± 5.1	0.55

CL: cervical lordosis, TK: thoracic kyphosis, LL: lumbar lordosis, PI: pelvic incidence, PT: pelvic tilt, SVA: sagittal vertical axis.

**Table 3 jcm-10-03182-t003:** The distribution of each factor at the different levels of conus medullaris.

	Total Number of Cases	T11-12	T12	T12-L1	L1	L1-2	L2	*p*
Height								
≤150 cm	51	1	4	22	19	5	0	0.67
150 to 175 cm	530	2	37	167	248	68	8	
>175 cm	48	0	5	15	21	6	1	
PI								
<30°	4	0	0	2	1	1	0	0.045
30–45°	139	0	7	41	69	19	3	
45–60°	330	1	22	96	154	50	6	
>60°	156	2	17	64	64	9	0	
PT								
<20°	472	1	29	149	226	61	6	0.12
20–30°	127	2	14	40	54	14	3	
>30°	30	0	3	15	8	4	0	

Each number shows the number of cases. Statistical analysis was performed by using a Chi-square test.

## Data Availability

The data of this study are available from the corresponding authors upon request.

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
