# Peer review of "The Level of Conus Medullaris in 629 Healthy Japanese Individuals"

_jcm, 2021, doi:10.3390/jcm10143182_

Round 1
Reviewer 1 Report
Congratulations on a well-conducted study.
It is readable and interesting.
I've added line# and comments below:
Line 50->
In the M and M section you excluded according to certain criteria. Did you diagnose/encounter any asymptomatic anomalies (e.q. tethered cord or other anomaly) within the included cohort or where asymptomatic anomalies excluded due to the causes mentioned in line55-57. If cases with asymptomatic findings where included, did they represent outliers regarding conus level??
Line 73-74:
Regarding the radiographic measurements. Did you apply a predefined protocol for the measurements. Please define. I'm not familiar with the osiris software and upon a brief google search I've not been able to determine exactly how this software applies the measures (how does it make the different measurements, is it automated). A short description would be appreciated in case of study reproducibility.
Line 75:
Multiple raters appear to have performed the measurements. Was it single measurements by random raters or did you evaluate intra and inter-rater reliability?
Line 89:
12.56%, please edit to 12.6% which makes the total =100%
Line 91:
"physics". I'm not native american english speaking but I think "physique" is the appropriate term, otherwise I will leave it up to the editing/proof-reading team of the journal.
Line 124-125:
You find a correlation between the PI end conus level but is that sufficient to determine it as a "key driver"? If evidence of causality exist, please add reference(s).
Line 146:
"length of spinal column / height". I can't find spinal column length measures anywhere in the article. How did you measure that? As the sitting height? Some find it to be better representative of spinal height than standing body height. If that is how you determined this measure please add this measure to table 1. Or did you measure the spinal height in the standing X-rays, then please add this parameter to table 2 containing the radiographic parameters.
Author Response
Reviewer 1
Congratulations on a well-conducted study.
It is readable and interesting.
I've added line# and comments below:
Line 50->
In the M and M section you excluded according to certain criteria. Did you diagnose/encounter any asymptomatic anomalies (e.q. tethered cord or other anomaly) within the included cohort or where asymptomatic anomalies excluded due to the causes mentioned in line55-57. If cases with asymptomatic findings where included, did they represent outliers regarding conus level??
Answer: No intradural anomalies including spinal cord tumor or tethered cord were found in this cohort. Lumbosacral transitional anomalies were found 21 cases, which were excluded from the current study. Although it is uncertain whether these anomalies affect the level of the conus, these cases were not included in this study, just as in our other studies [1,2] using the same cohort. As the reviewer commented, however, it is necessary to examine the level of conus in these cases of transitional vertebra in the future. We added this in the limitation (line 233-239).
References
- Yukawa Y, Matsumoto T, Kollor H et al. Local Sagittal Alignment of the Lumbar Spine and Range of Motion in 627 Asymptomatic Subjects: Age-Related Changes and Sex-Based Differences. Asian spine j 2019; 13: 663-671 DOI: 10.31616/asj.2018.0187
- Yukawa Y, Kato F, Suda K et al. Normative data for parameters of sagittal spinal alignment in healthy subjects: an analysis of gender specific differences and changes with aging in 626 asymptomatic individuals. European spine journal 2018; 27: 426-432 DOI: 10.1007/s00586-016-4807-7
Line 73-74:
Regarding the radiographic measurements. Did you apply a predefined protocol for the measurements. Please define. I'm not familiar with the osiris software and upon a brief google search I've not been able to determine exactly how this software applies the measures (how does it make the different measurements, is it automated). A short description would be appreciated in case of study reproducibility.
Answer: Each parameter was “manually” measured by experienced radiation technologists by using Osiris4. This Osiris4 is a software that allows manually measuring the length, angle and the area of the images obtained via the Picture Archiving and Communication Systems (PACS) (line 95-97).
Line 75:
Multiple raters appear to have performed the measurements. Was it single measurements by random raters or did you evaluate intra and inter-rater reliability?
Answer: This was single measurements by random raters (line 94).
Line 89:
12.56%, please edit to 12.6% which makes the total =100%
Answer: We changed it to 12.6% (line 113).
Line 91:
"physics". I'm not native american english speaking but I think "physique" is the appropriate term, otherwise I will leave it up to the editing/proof-reading team of the journal.
Answer: We changed it to physique (line 115).
Line 124-125:
You find a correlation between the PI end conus level but is that sufficient to determine it as a "key driver"? If evidence of causality exist, please add reference(s).
Answer: As far as we know, there is no paper showing the relationship between PI and the conus level due to the lack of studies investigating the conus level by using both lumbar MRI and X-rays.
For this reason, the current results will need to be verified in other future studies (line 200-203).
Line 146:
"length of spinal column / height". I can't find spinal column length measures anywhere in the article. How did you measure that? As the sitting height? Some find it to be better representative of spinal height than standing body height. If that is how you determined this measure please add this measure to table 1. Or did you measure the spinal height in the standing X-rays, then please add this parameter to table 2 containing the radiographic parameters.
Answer: We apologize for this confusing description. The spinal height was not measured in the current study. In cases with a large PI, LL generally tend to be large. Considering the current results, the spinal cord length may have less variation, resulting in the spinal cord being located more on the cranial side since it passes through a winding spinal canal (line 210-212). This is just our hypothesis and needs to be verified in the future.

Reviewer 2 Report
This research can provide some interesting evidence to readers about the lumbar spine pain and the related factors. In this regard, the authors should provide more details about the inclusion criteria and the risk factors related to the pathophysiology of spinal specific or non-specific pain generators. Also the authors should provide the statistical considerations and details. The consideration of pathophysiology of spinal pathology should be considered from the beginning of research design.
Author Response
Reviewer 2
This research can provide some interesting evidence to readers about the lumbar spine pain and the related factors. In this regard, the authors should provide more details about the inclusion criteria and the risk factors related to the pathophysiology of spinal specific or non-specific pain generators. Also the authors should provide the statistical considerations and details. The consideration of pathophysiology of spinal pathology should be considered from the beginning of research design.
Answer: With respect to pain, we took a visual analogue scale (VAS) measurement of low back, buttock and leg pain before deciding on inclusion of patients in this study, and excluded cases with severe pain anywhere above 80 mm as cases with severe pain (line 67-69). As a limitation of the study, we couldn’t compare spinal alignment and the level of conus medullaris by degrees of pain, although the degrees of pain might affect the results. Future detailed studies assessing the pain are needed (line 239-241). With respect to inclusion and exclusion criteria, we excluded cases with previous medical history of vertebral fracture, spinal infection, rheumatoid arthritis, autoimmune diseases or chronic renal failure. In contrast, we included cases with diabetes mellitus or smoking history. We also added the details of statistical analysis (line 72-75). Thank you for your valuable comments.

Reviewer 3 Report
Introduction
The rationale of the study is not made clear precisely. What is the clinical relevance, that the readership of JCM needs to know?
I.e. some surgical or analytical procedures (like LP) rely on the correct position of the conus medullaris.
M&M
The exclusion of severe back pain clearly reduces the applicability for the medical field.
So was the study design prospective? If so, please explicitly state. The ethical reason for an MRI in patients without abnormalities or pain and without a distinct clinical benefit from the outcome can be questioned. And they get an additional X-ray (full-length), how can this be ethically reasonable, especially without an individual (or necessary overall population) benefit?
Was the study registered?
Were the groups comparable in terms of gender, height and weight?
Did you do a one-way, two-way or three-way ANOVA? Did you assess the standard distribution beforehand?
Results
Please further state whether a high PI and PT lead to higher or lower conus medullaris levels. The same thing should be done with body height. Please validate whether the sex causes differences regarding the conus medullaris level.
Discussion
The second paragraph should go into the introduction.
Just the final paragraph belongs into the discussion. Please expand this section significantly.
Author Response
Reviewer 3
Introduction
The rationale of the study is not made clear precisely. What is the clinical relevance, that the readership of JCM needs to know?
I.e. some surgical or analytical procedures (like LP) rely on the correct position of the conus medullaris.
M&M
The exclusion of severe back pain clearly reduces the applicability for the medical field.
So was the study design prospective? If so, please explicitly state. The ethical reason for an MRI in patients without abnormalities or pain and without a distinct clinical benefit from the outcome can be questioned. And they get an additional X-ray (full-length), how can this be ethically reasonable, especially without an individual (or necessary overall population) benefit?
Answer: As part of a comprehensive medical examination, the study was conducted after consent was obtained from subjects who wanted spinal examinations. Participants were offered free feedback on findings from spine radiographs and MRIs, rather than monetary rewards, and the study was conducted with the approval of an ethics committee. We prospectively recruited the subjects using newspaper advertisements and posters in facilities having some sort of relationship with our hospital. The majority of the subjects were not patients at our hospital but relatively healthy residents of the area (line 56-62).
As the reviewer commented, back pain affect the result. We took a visual analogue scale (VAS) measurement of low back pain before deciding on inclusion of patients in this study, and excluded cases with severe pain anywhere above 80 mm as cases with severe pain, although there are several cut-off points for low back pain in the VAS (line 67-69). As a limitation of the study, we couldn’t compare spinal alignment and the levels of conus medullaris by degrees of pain, although these levels of pain might affect the results. Future detailed studies assessing the pain is needed (line 239-241). I added these points to the Method and Limitations sections.
Was the study registered?
Answer: Although this study was registered in the research database at Rosai Hospitals in Japan, it was not officially registered (line 62-63).
Were the groups comparable in terms of gender, height and weight?
Answer: Table 1 shows the relevant data. Regarding gender, there was no significant difference among the levels of conus medullaris (line 119).
Did you do a one-way, two-way or three-way ANOVA? Did you assess the standard distribution beforehand?
Answer: At first, we assessed the standard distribution of each parameter (age, height, weight, BMI and radiographical parameters) by using Kolmogorov-Smirnov test. After confirmation of normal distribution, we employed one-way ANOVA (post hoc Tukey) to investigate the differences for each parameter at the different conus medullaris levels (line 100-105).
Results
Please further state whether a high PI and PT lead to higher or lower conus medullaris levels. The same thing should be done with body height. Please validate whether the sex causes differences regarding the conus medullaris level.
Answer: We added the following Table 3 as the reviewer commented. The gender-related data were added in Table 1, although gender was not found to be a significant factor (p=0.48) among the different levels of conus medullaris.
Table 3. The distribution for each factor at the different levels of conus medullaris
|
|
Total number of cases |
T11-12 |
T12 |
T12-L1 |
L1 |
L1-2 |
L2 |
p |
|
Height |
||||||||
|
≤150 cm |
51 |
1 |
4 |
22 |
19 |
5 |
0 |
0.67 |
|
150 to 175 cm |
530 |
2 |
37 |
167 |
248 |
68 |
8 |
|
|
>175 cm |
48 |
0 |
5 |
15 |
21 |
6 |
1 |
|
|
PI |
||||||||
|
< 30° |
4 |
0 |
0 |
2 |
1 |
1 |
0 |
0.045 |
|
30-45° |
139 |
0 |
7 |
41 |
69 |
19 |
3 |
|
|
45-60° |
330 |
1 |
22 |
96 |
154 |
50 |
6 |
|
|
> 60° |
156 |
2 |
17 |
64 |
64 |
9 |
0 |
|
|
PT |
||||||||
|
< 20° |
472 |
1 |
29 |
149 |
226 |
61 |
6 |
0.12 |
|
20-30° |
127 |
2 |
14 |
40 |
54 |
14 |
3 |
|
|
> 30° |
30 |
0 |
3 |
15 |
8 |
4 |
0 |
|
Each number shows number of cases. Statistical analysis was performed by using a Chi-square test.
Discussion
The second paragraph should go into the introduction.
Just the final paragraph belongs into the discussion. Please expand this section significantly.
Answer: We moved the second paragraph into the introduction (line 30-35) and changed the Discussion section (line 213-226). Thank you for all of your valuable comments.
Round 2
Reviewer 3 Report
Just to clarify, the patients wanted an X-ray after being informed that after all radiation can have negative effects?
If the study was not registered beforehand, this should be done at least now.
The new section regarding the PI offers helpful insights.
Author Response
Just to clarify, the patients wanted an X-ray after being informed that after all radiation can have negative effects?
Answer: All of the included volunteers understood negative effects of radiation exposure and agreed to undergo X-ray examination. We added this in the method (line 59-60).
If the study was not registered beforehand, this should be done at least now.
Answer: We agree with the reviewer’s suggestion. After your previous review, we discussed to register this study in an official registration system among ourselves.
The new section regarding the PI offers helpful insights.
Answer: We appreciate your valuable comments.